# Adjuvant use of melatonin for pain management in endometriosis-associated pelvic pain—A randomized double-blinded, placebo-controlled trial

**Lisa Söderman**[1]*, **Ylva Böttiger**[2], **Måns Edlund**[3], **Hans Järnbert-Pettersson**[1], **Lena Marions**[1]

**1** Department of Clinical Science and Education, Södersjukhuset, Karolinska Institutet, Stockholm, Sweden, **2** Department of Biomedical and Clinical Sciences, Linköping University, Linköping, Sweden, **3** KBH, Department of Womens and Childrens Health, Karolinska Institutet, Stockholm, Sweden

* lisa.soderman@ki.se

## Abstract

Considering the pharmacological treatment options for endometriosis-associated pain are confined to hormonal therapy and analgesics, we studied the analgesic effect of 20 mg melatonin as an adjuvant therapy in women with endometriosis-associated pain. This randomized double-blinded, placebo-controlled trial was conducted at the Research Center for Womens' Health at Södersjukhuset, a university hospital in Stockholm, Sweden. Forty women from 18 to 50 years of age with endometriosis and severe dysmenorrhea with or without chronic pelvic pain were given 20 mg Melatonin or placebo orally daily for two consecutive menstrual cycles or months. The level of pain was recorded daily on the 11-point numeric rating scale, a difference of 1.3 units was considered clinically significant. Clincaltrials.gov nr NCT03782740. Sixteen participants completed the study in the placebo group and 18 in the melatonin group. The difference in endometriosis-associated pain between the groups showed to be non-significant statistically as well as clinically, 2.9 (SD 1.9) in the melatonin group and 3.3 (SD 2.0) in the placebo group, p = 0.45. This randomized, double-blinded, placebo-controlled trial could not show that 20 mg of melatonin given orally at bedtime had better analgesic effect on endometriosis-associated pain compared with placebo. No adverse effects were observed.

## Introduction

Endometriosis is estimated to affect approximately 10% of women of reproductive age [1] defined by presence of ectopic endometrial-like tissue. It is a chronic systemic inflammatory disease. The most common symptoms are severe dysmenorrhea and chronic pelvic pain which are disabling to many women and, in many cases, refractory to treatment. Available treatments for endometriosis-associated pain (EAPP) consist of analgesics, hormonal therapy and sometimes surgery.

**Data Availability Statement:** Due to the study design the data is not anonymized but pseudonymized. The raw data includes identifiers

and sensitive patient information. The data underlying the results presented in the study are available for researchers who meet the criteria for access to confidential data from SND (Swedish National Data Service) at https://doi.org/10.48723/875f-ma81.

**Funding:** This study was supported by grants provided by the Stockholm County Council (ALF project 20180306), AFA insurance (project number 170157) and Karolinska Institutet fund for endometriosis research. The funders had no role in study design, data collection and analysis, decision to publish, or preparation of the manuscript.

**Competing interests:** Competing Interests: LS, LM, YB, HJP have no conflict of interest. ME reports employment and stock ownership with SOBI AB, and previously with Vifor Pharma AB. This does not alter our adherence to PLOS ONE policies on sharing data and materials.

Melatonin could be a treatment option due to its anti-inflammatory [2], analgesic [3], anti-estrogenic properties [4] as well as its favorable safety profile [5]. Melatonin has been shown to impair the cellular invasion, migration and proliferation of endometriosis [6–10].

We recently explored the analgesic effect of 10 mg melatonin daily in women suffering from severe dysmenorrhea, however we could not show this regimen to be superior to placebo [11].

To our knowledge, there is one previous study assessing the effect of melatonin showing 10 mg melatonin daily to reduce EAPP [12] in a clinically relevant manner. There is a need for clinical studies to further examine the effects in women with EAPP.

We conducted our study with 20 mg melatonin daily to further assess melatonin as a possible adjunct treatment in EAPP.

## Materials and methods

We conducted this randomized, double-blinded, parallel placebo-controlled trial at Södersjukhuset, a hospital in Stockholm, Sweden. Participants were recruited between August 2019 and March 2021. Prior to enrolment, a written informed consent was obtained from the participants. The principles expressed in the Declaration of Helsinki were respected in conducting the trial. The trial was approved by The Regional Ethical Review Board at Karolinska Institutet (2017/1177-21/2). Registration number at Clincaltrials.gov NCT03782740. The first participant was enrolled 21August 2019, last patient last visit was 27 June 2021.

Call for participation was advertised on posters in the hospital, in gynecological outpatient clinics, in maternity care outpatient clinics, and on social media. We recruited women who rated their dysmenorrhea 7 or higher on a numeric rating scale (NRS) during the most painful day or a mean NRS of 3 or more for seven days with endometriosis diagnosed through laparoscopy, ultrasound or MRI. All participants were aged 18–50, in good general health and speaking and understanding Swedish. Initial recruitment included only women without any hormonal treatment with regular menstrual bleedings. The inclusion criteria was expanded to women with hormonal treatment, with or without amenorrhea, due to recruitment difficulties. Screening was made by phone followed by one observational menstrual cycle or 30-day period, during this period of time pain was recorded daily, and evaluated prior to inclusion. The first patients enrolled came for a visit with one of the doctors in charge of the trial, at the clinics Research Center for Womens' Health for inclusion. The visit included medical history and a pregnancy test. Due to Covid-19 the study was converted entirely to remote visits. The inclusion visit was substituted with a video call for identification with id-card, after randomization the study drugs were sent by registered mail or delivered personally to participants living in Stockholm, a pregnancy test and a consent form were included in the consignment. A picture of the taken pregnancy test was returned prior to commencing with the study drug and the signed form of consent was returned by mail. Exclusion criteria were smoking, pregnancy, prior or current liver or kidney disease, ongoing use of melatonin, alteration of any medication during the last three months, regular use of opioids.

After inclusion, participants were randomized to 20 mg melatonin or placebo, each dose identical and dispersed in four capsules of 5 mg melatonin or placebo (both manufactured for the trial by APL, Stockholm, Sweden). The study drug was taken at bedtime daily for two consecutive menstrual cycles with start on the first day of menstrual bleeding or 60 consecutive days for amenorrhoeic participants. If needed, the participants continued with their usual pain medication regime, for the three menstrual cycles/ months of the study. The study period was one observational cycle/month followed by two interventional cycles/months, i.e. three menstrual cycles/months in total.

The primary outcome was EAPP recorded daily. The secondary outcomes were use of analgesics, dysuria, dyschezia, dyspareunia, days of pain, absenteeism, quality of life, pain catastrophization and sleep.

Daily assessments of pain (using NRS), use of analgesics (with specified dosage and number of tabelts), absenteeism and bleeding using a pictorial blood loss assessment chart (PBAC) were made at bedtime using an online questionnaire sent by email every day. Recording of potential adverse events was included in the questionnaire. General experience of the study drug was evaluated at completion of the study.

Quality of life was assessed with Endometriosis Health Profile-30 (EHP-30), chronic pain was assessed with the Pain Catastrophizing Scale and sleep was assessed with Insomnia sleep index (ISI) all done with online questionnaires during the first and last cycle/month respectively.

The study data was collected and then managed using REDCap electronic data capture tools (9.5.9 Vanderbuilt University, Nashville TN, USA) hosted at Karolinska Institutet.

Participants were randomized consecutively in order of inclusion. The manufacturer of the study drug provided consecutively numbered drug containers and made the randomization in blocks of 4. The randomization key was retrieved and opened after the last participant had completed the study asserting double blinding.

## Statistical analysis

To detect a clinically significant reduction of NRS of 1.3 units [13] and an assumed standard deviation (SD) of 1.2 with a power of 80%, and a 2-sided alpha value of 0.05, 15 participants in each group were needed. Like a similar study [12], we included 20 participants in each group, in total 40 participants, to compensate for those lost to follow up (Fig 1). The calculation was done with sample-size.net [14] using independent t-test.

We used mixed models analysis to study if there were any differences between the groups over all time points for each of the five continuous outcomes recorded daily (EAPP, amount of ingested analgesic, dysuria, dyschezia and dyspareunia) [15].

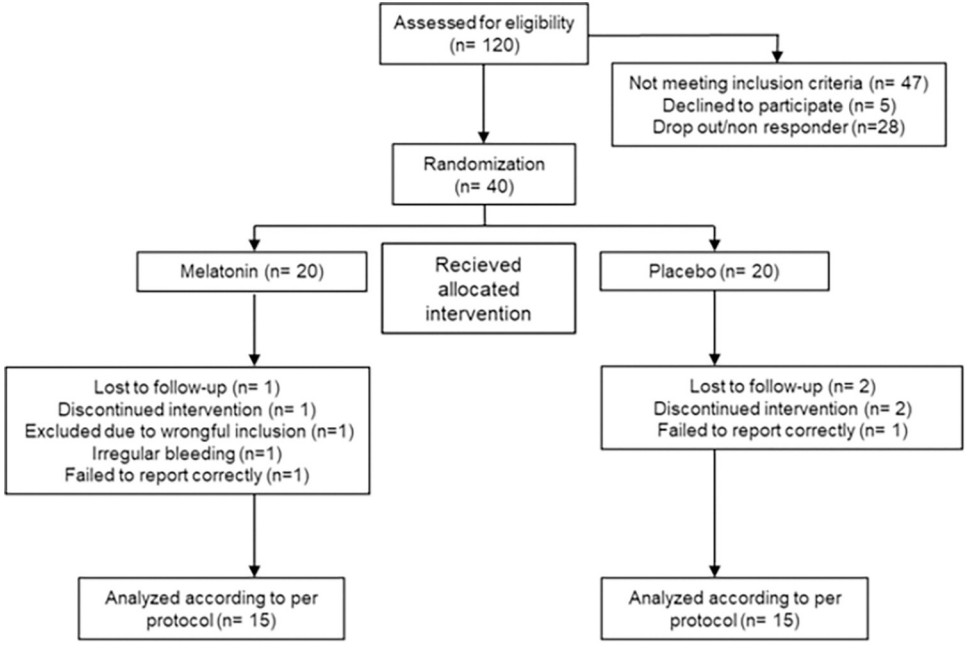

**Fig 1. CONSORT flow chart.**

Covariance structure AR(1) with random intercept, without random slope proved to be the best fit for all outcomes. We used a model with time and treatment as fixed factors. To study if the effect between the treatments differed over time, we added an interaction between time and treatment. In addition, to study if weight and hormonal therapy would adjust the treatment effect these factors were added one at a time to the model with treatment and time and thereafter interactions between weight and treatment as well as for hormonal therapy and treatment were tested one at a time. We averaged the daily recorded values to generate one value per week, keeping the menstrual cycle as an independent variable. The first day was excluded from inference since the study drug was ingested at bedtime.

Data was analysed according to the randomized treatment assignments, for patients who fulfilled the inclusion and exclusion criteria (intention-to-treat), as well as according to the per-protocol. Because the results were similar, we only present the ITT analysis.

Independent t-tests were used to compare continuous data in baseline characteristics, mean days of dysmenorrhea, days of using analgesics, EHP-30, pain catastrophizing scale and insomnia sleep index. Categorical data was assessed with chi square or Fisher's exact test. Acceptability was compared with Fisher's exact test. A two-tailed p-value of less than 0.05 was considered to indicate statistical significance. SPSS version 26 (SPSS, Chicago, IL) was used for data analyses.

## Results

We screened 120 volunteers to include 40 participants, 20 in each group (Fig 1). Sixteen participants completed the study in the placebo group and 18 in the melatonin group. One participant in the melatonin group was lost to follow up and one chose to discontinue due to restless legs (a known side effect). In the placebo group, two were lost to follow up and two discontinued, one due to abdominal pain and one due to forgetting to take the study drug. Main characteristics of the study population are presented in Table 1. The outcomes are presented as means and SD of the screening cycle.

All variables were similar between the groups except for dyspareunia. In the placebo group endometriosis had been diagnosed in 17 (85%) participants through laparoscopy, 2 (10%) with ultrasound and one (5%) through laparotomy. In the melatonin group 13 (65%) had been diagnosed through laparoscopy, 5 (25%) with ultrasound, 1(5%) with MRI and one (5%) with symptomatology (the latter was wrongfully included in the study). In the placebo group 16 (80%) participants reported no comorbidities, one (5%) participant had migraine, one (5%) had fibromyalgia and one (5%) had a hiatal hernia. In the melatonin group 15 (75%) participants reported no comorbidities, one (5%) had hypothyroidism, one (5%) had treatment for high blood pressure, one (5%) had had a gastric sleeve done and PCOS. One (5%) in each group had had a gastric by-pass surgery. Two (20%) participants in the melatonin group took antidepressants.

No significant differences in primary or secondary outcomes were seen between the treatment groups (Table 2).

Adjusting for weight or for hormonal therapy did not affect the results. A sensitivity analysis stratified by hormonal treatment showed no differences between the groups (S1 Table). No statistically significant interaction between time and treatment group was seen, which imply similar output between the groups for all time points. The level of insomnia was similar between the treatment groups (Table 3). Four (25%) participants in the placebo group had clinical insomnia (ISI >14), 3 (17%) in the melatonin group. Quality of life, assessed with EHP as well as the level of PCS was similar between the groups (Table 3). No difference in the mean number of days per cycle with EAPP, which were 22.6 (SD 9.17) and 21.0 (SD 8.14) or mean

**Table 1. Baseline characteristics.**

| Mean (SD) | Placebo n = 20 | Melatonin n = 20 | p-value |
|---|---|---|---|
| Age | 34.20 (7.68) | 35.90 (6.61) | .458 |
| Length, cm | 168.15 (6.01) | 167.15 (6.34) | .612 |
| Weight, kg | 68.80 (6.80) | 72.15 (15.46) | .381 |
| Number of pregnancies | 1.45 (1.73) | 1.20 (1.47) | .626 |
| Number of deliveries | .70 (1.22) | .70 (1.03) | 1.0 |
| Number of miscarriages | .30 (.80) | .35 (.80) | .846 |
| **Contraceptives n (%)** | | | .462 |
| None | 6 (30%) | 5 (25%) | |
| Condom | 3 (15%) | 7 (35%) | |
| Progestin pill (eg desogestrel) | 1[a] (5%) | 0 | |
| Hormonal IUS 52 mg LNG | 4[a] (20%) | 4 (20%) | |
| Hormonal IUS 19,5 mg LNG | 1 (5%) | 0 | |
| COCP | 1 (5%) | 0 | |
| Sterilization | 0 | 1 (5%) | |
| Missing data | 4 (20%) | 3 (15%) | |
| **Use of hormonal therapy n (%)** | | | .288 |
| Without hormonal therapy | 13 (65%) | 16 (80%) | |
| With hormonal therapy | 7 (35%) | 4 (20%) | |
| **Mean (SD)** | | | |
| EAPP mean | 3.63 (1.87) | 2.87 (1.86) | .202 |
| Analgesics in mg, mean | 825.31 (981.11) | 661.90 (776.73) | .563 |
| Dysuria, mean | 1.60 (1.77) | 1.56 (2.13) | .951 |
| Dyschezia, mean (n 20/19) | 2.14 (2.09) | 1.58 (1.98) | .398 |
| Dyspareunia, mean (n 16/14) | 2.08 (2.47) | .58 (.93) | .041 |
| **Mean (SD)** | | | |
| Insomnia severity index | 13.75 (6.11) | 12.35 (6.01) | .470 |
| EHP—pain | 48.30 (19.96) | 48.52 (22.58) | .973 |
| EHP—control | 61.88 (23.93) | 60.21 (22.92) | .823 |
| EHP—emo | 48.13 (21.65) | 49.17 (19.99) | .875 |
| EHP—social | 53.13 (21.51) | 49.06 (28.26) | .612 |
| EHP—self esteem | 61.25 (27.87) | 62.08 (21.37) | .916 |
| Pain catastrophizing scale | 25.25 (9.38) | 27.55 (10.33) | .466 |

[a] One participant in the placebo group had both hormonal IUS and Dienogest. LNG = levonorgestrel. COCP = combined oral contraceptive pill.

number of days with analgesics, which were 10.8 (SD 13.18) and 10.4 (SD 9.20) in the placebo and melatonin group respectively. Out of the analgesics taken during the study period acetaminophen made up 48%, NSAIDs 46%, codeine 5% and 1% other opioids. In the placebo group there were 4 whole days of absenteeism and 6 half days, in the melatonin group there was one half day of absenteeism, but 68 days off compared to 41 in the placebo group.

When assessing acceptability at the end of the study period 11 (61%) in the melatonin group had a good experience with the drug, 1 (6%) had a bad experience and 6 (33%) did not know. In the placebo group the reports showed 9 (56%), 3 (19%) and 4 (25%) respectively. 11 (61%) and 10 (62.5%) would recommend the study drug to a friend in the melatonin group and placebo group respectively. No statistically significant differences were seen between the treatment groups in regard to having had a good or bad experience with the drug or recommending it to a friend.

**Table 2. Treatment effect on the outcomes during the study period.**

|  | Treatment group | n | Adjusted mean (SD) | Adjusted mean difference | Confidence Interval 95% | p-value |
|---|---|---|---|---|---|---|
| Endometriosis-associated pain | Placebo | 20 | 3.3 (2.0) | .4 | -.7 to 1.4 | .446 |
|  | Melatonin | 20 | 2.9 (1.9) |  |  |  |
| Analgesics, mg | Placebo | 20 | 505.5 (762.4) | -136.7 | -571.7 to 298.3 | .529 |
|  | Melatonin | 20 | 642.2 (915.9) |  |  |  |
| Dysuria | Placebo | 20 | 1.1 (1.7) | .0 | -1.0 to 1.1 | .930 |
|  | Melatonin | 20 | 1.1 (1.8) |  |  |  |
| Dyschezia | Placebo | 20 | 1.7 (2.0) | .7 | -.5 to 1.7 | .263 |
|  | Melatonin | 20 | 1.0 (1.7) |  |  |  |
| Dyspareunia | Placebo | 18 | 1.1 (1.8) | .4 | -.7 to 1.4 | .499 |
|  | Melatonin | 17 | .7 (1.2) |  |  |  |

Analyzed with linear mixed models.

## Discussion

This randomized double-blinded placebo-controlled study could not show that 20 mg melatonin ingested at bedtime reduced endometriosis-associated pain, in contrast to a previous study [12] that showed a significant reduction of menstrual and chronic pain in women with endometriosis with a dosage of 10 mg melatonin daily. Previous studies, both animal [16] and human [17], have shown the analgesic effect of melatonin to be dose dependent. We could not show the same effect, perhaps the fact that our study population had lower pain scores overall had an impact. Pain reduction in the higher range of NRS might be more noticeable than in the lower range. In the Schwertner study participants were recruited with chronic pelvic pain and/or dyspareunia and had the endometriosis diagnosis confirmed through laparoscopy by one of the authors. Participants with different stages and different subtypes of endometriosis were included and recruited from a medical setting. Smokers were included. Their mean level of dysmenorrhea at baseline was 8 on the visual analogue scale in the placebo group and 7.32

**Table 3. Treatment effect of the outcomes analyzed at the end of the study.**

| End of treatment outcomes |  | Treatment | n | Mean (SD) | Mean difference | 95% Confidence intervals | p-value |
|---|---|---|---|---|---|---|---|
| Insomnia severity index |  | Placebo | 16 | 9.6 (4.7) | -0.2 | -3.6 to 3.2 | 0.902 |
|  |  | Melatonin | 18 | 9.8 (5.0) |  |  |  |
| Endometriosis Health Profile | Pain | Placebo | 16 | 31.0 (18.3) | -3.9 | -17.3 to 9.6 | 0.561 |
|  |  | Melatonin | 18 | 34.9 (20.0) |  |  |  |
|  | Control & powerlessness | Placebo | 16 | 44.3 (20.4) | -0.2 | -15.2 to 14.9 | 0.981 |
|  |  | Melatonin | 18 | 44.5 (22.5) |  |  |  |
|  | Emotional & well being | Placebo | 16 | 39.1 (15.36) | 2.0 | -10.5 to 14.5 | 0.744 |
|  |  | Melatonin | 18 | 37.0 (19.9) |  |  |  |
|  | Social support | Placebo | 16 | 41.0 (29.0) | 2.1 | -16.6 to 20.9 | 0.819 |
|  |  | Melatonin | 18 | 38.9 (24.8) |  |  |  |
|  | Self image | Placebo | 16 | 47.4 (25.4) | -2.1 | -19.2 to 14.9 | 0.800 |
|  |  | Melatonin | 18 | 49.5 (23.5) |  |  |  |
| Pain catastrophization |  | Placebo | 16 | 23.2 (10.9) | -0.3 | -7.9 to 7.4 | 0.946 |
|  |  | Melatonin | 18 | 23.4 (11.0) |  |  |  |

Independent t-test was used to evaluate sleep, quality of life and pain catastrophization. The participants answered to questionnaire on day 21 in the last treatment cycle.

in the melatonin group, the level of chronic pain where the week of menstruation was excluded was 6.89 in the placebo group and 6.46 in the melatonin group [12]. The mean pain levels in dysuria, dyschezia and dyspareunia are very low in our study, probably too low to detect an improvement. The methodology in both studies was similar regarding timing of melatonin administration, recording of outcome measures and statistical analysis.

Studies have shown that melatonin inhibits the estrogen-driven epithelial cell migration, invasion and epithelial-mesenchymal transition (EMT) [6] as well as proliferation [18] of endometriotic cells, stipulating melatonin may influence development and progression of endometriosis. The endometriotic lesion itself produces estradiol and prostaglandin through a positive feedback loop. The estradiol promotes survival, proliferation, and inflammation in the poorly differentiated endometrial stromal cells. The prostaglandins cause inflammation and endometriosis associated pelvic pain [19]. This chronic inflammation can potentially lead to peripheral nerve stimulation and sensitization amplifying the local inflammatory response and generation of pain [20]. Central sensitization is initiated by peripheral sensitization and can become autonomous and generate pain without peripheral noxious stimulus [21].

A significantly higher level of protein oxidative stress markers has been shown in the peritoneal fluid of women with endometriosis compared to controls, with a positive correlation between the level of advanced oxygen protein products and the pelvic pain symptom scores [22]. Melatonin is a well-documented scavenger of free radicals. The anti-oxidative effect is directly related to its concentration. At higher concentrations, there are more molecules of the antioxidant available to quench free radicals thereby lowering oxidative damage and related diseases [2]. It can also stimulate antioxidant enzymes in different tissues [23]. Animal studies have shown that melatonin reduces the size of endometriotic implants in rats [7, 8, 24, 25] and Cetinkaya et al suggested a positive dose dependent relationship to the size reduction [26] as well.

The analgesic effects of melatonin are not fully understood and have been attributed to its anti-oxidative properties, but it is now evident that naloxone inhibits the antinociceptive effect of melatonin concluding the involvement of the opioid system, perhaps via melatonin receptors present in the spinal cord and in the brain [3].

Our regime was fixed to taking the study drug at bedtime to mimic the endogenous production of melatonin, considering the short half-life of melatonin in humans (20–40 minutes) [27] we most likely did not evaluate the analgesic effect of melatonin on acute pain.

The given dose and/or mode of administration might not have been sufficient to render any anti-inflammatory effect, to reduce the size of endometrial lesions and to reduce inflammatory pain. A mode of administration with longer duration such as transdermal application, assessed with serum levels of melatonin as well as measuring its clinical effect on pain, could provide information on how to treat endometriosis with melatonin. We could not show any differences between the groups in quality of life or pain catastrophization which were included to assess the more chronic effect of pain.

Plasma concentrations of melatonin were not evaluated, which would have provided valuable information, since bioavailability is low at 15% and is associated with high inter-subject variability [27].

The strengths of the study include the comparison of melatonin and placebo in a double blinded, parallel placebo-controlled trial, the low level of dropouts and missing data and the low risk for recollection bias with a daily questionnaire. There was a high rate of adherence to the study drug, no participant had lower than 82% adherence. The study design offers high internal validity. The trial was conducted according to the CONSORT guidelines [28].

The study limitations involve study design as well as method. By including only self-selected non-smoking, Swedish-speaking women in good health our study population is lacking in

diversity with a possibly low external validity due to selection bias. By underestimating the standard deviation of the primary outcome the study might be underpowered, as the effect of 1.3 units is included in the confidence interval, a type II error cannot be excluded and the result could be inconclusive [29]. By including different modes of diagnosis of endometriosis and only self-reported diagnosis misclassification bias may have been introduced which could have affected the results.

## Conclusion

Our study could not show that 20 mg of melatonin given orally at bedtime during two menstrual cycles/ two months had better analgesic effect on endometriosis-associated pain as compared with placebo. We studied a higher dose melatonin than previously studied for EAPP, however no major adverse effects were observed. The mean levels of pain in our study might have been too low to detect any effect compared with placebo. Further studies with larger study populations may be of use to this group of patients suffering from chronic pain and limited treatment options.

## Supporting information

**S1 Checklist. CONSORT 2010 checklist of information to include when reporting a randomised trial\*.**
(DOC)

**S1 Table. Sensitivity analysis.** Stratification by hormonal treatment end weight, respectively.
(XLSX)

**S1 File.**
(DOCX)

**S2 File.**
(XLSX)

## Author Contributions

**Conceptualization:** Lena Marions.

**Data curation:** Lisa Söderman.

**Formal analysis:** Lisa Söderman, Hans Järnbert-Pettersson.

**Investigation:** Lisa Söderman.

**Methodology:** Lisa Söderman, Ylva Böttiger, Måns Edlund.

**Project administration:** Lisa Söderman.

**Resources:** Lisa Söderman, Lena Marions.

**Software:** Lisa Söderman.

**Supervision:** Lena Marions.

**Writing – original draft:** Lisa Söderman.

**Writing – review & editing:** Lisa Söderman, Ylva Böttiger, Måns Edlund.

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
