## [Decision Letter · Decision Letter 0]

7 Feb 2023

PONE-D-22-34783Adjuvant use of melatonin for pain management in endometriosis-associated pelvic pain - a randomized double-blinded, placebo-controlled trialPLOS ONE

Dear Dr. Söderman,

Thank you for submitting your manuscript to PLOS ONE. After careful consideration, we feel that it has merit but does not fully meet PLOS ONE’s publication criteria as it currently stands. Therefore, we invite you to submit a revised version of the manuscript that addresses the points raised during the review process.

ACADEMIC EDITOR:Please respond to all reviewers comments clearly point by point

We look forward to receiving your revised manuscript.

Kind regards,

Ahmed Mohamed Maged, MD

Academic Editor

PLOS ONE

Journal Requirements:

- 10.1007/s00228-021-03234-6. Epub 2021 Oct 20. 

In your revision ensure you cite all your sources (including your own works), and quote or rephrase any duplicated text outside the methods section. Further consideration is dependent on these concerns being addressed.

"LS, LM, YB, HJP have no conflict of interest. ME reports employment and stock ownership with SOBI AB, and previously with Vifor Pharma AB."

Additional Editor Comments:

Please respond to all reviewers comments

Reviewers' comments:

Reviewer's Responses to Questions

**Comments to the Author**

1. Is the manuscript technically sound, and do the data support the conclusions?

Reviewer #1: Yes

Reviewer #2: Yes

Reviewer #3: Yes

2. Has the statistical analysis been performed appropriately and rigorously? 

Reviewer #1: Yes

Reviewer #2: Yes

Reviewer #3: Yes

3. Have the authors made all data underlying the findings in their manuscript fully available?

Reviewer #1: No

Reviewer #2: No

Reviewer #3: Yes

4. Is the manuscript presented in an intelligible fashion and written in standard English?

Reviewer #1: Yes

Reviewer #2: Yes

Reviewer #3: Yes

5. Review Comments to the Author

Reviewer #1: A randomized-controlled clinical trial compared the level of pain between melatonin treatment and placebo in women with endometriosis-associated pain. No statistical differences in pain levels between the arms were observed.

Minor revisions:

1- Line 113: State the statistical testing method that achieves 80% power. If an estimate of the standard deviation was used in the power calculation, state it.

2- Line 127: The standard statistical term for average is mean.

3- Table 1: Indicate if the summarized values in the table are means and standard deviations. If data is normally distributed, means and standard deviations are appropriate. However, if the data is non-normally distributed median, first and third quartiles are typically provided.

4- Line 148: Indicate the statistical testing method(s) used to conclude that the variables in table 1 were similar/different when comparing the groups.

5- Line 149-156: In addition to the frequencies, provide the corresponding percentages.

6- Line 161: Consider rephrasing the following sentence. Both the terms differences and similar were used which is confusing to the reader. “No statistically significant interaction between time and treatment group was seen, which imply that differences between the groups were similar for all time points.”

7- Line 163: State the percent with insomnia in each arm.

8- Define the abbreviation SD at its first occurrence.

Reviewer #2: This is a well thought out experiment. I agree with the authors that the context of the experiment allows for high internal validity. It is clear that the authors are aware of both the strengths and the weaknesses of the experiment. I think implementing a stratified random sampling to account for previous hormone therapy users and any other important confounders, ie, the number of amenorrhea patients recruited after the exclusion criteria was relaxed, would benefit the analyses and interpretation of the data. It also would be helpful to know when the patients in both arms conducted their self-assessments, since it is stated that the half life of melatonin is somewhere between 20 and 60 minutes. For that matter, knowing the within-patient variance of documentation of self-assessments would be beneficial to the strength of the study. The authors need to better define and describe the interaction terms used in the regression models. And disclosure of the model equation, including estimates and SEs, should be done before final acceptance. It is my judgment that this experiment adds value to the scientific conversation regarding the association between melatonin and EAPP.

Reviewer #3: Thank you for giving me the opportunity to review this interesting RCT "Adjuvant use of melatonin for pain management in endometriosis-associated pelvic pain".

However, I have some comments and recommendations:

Title:- Is appropriate for the content of the article.

Abstract:- Represents a suitable summary of the study.

Introduction:- Emphasize the mechanism of action of melatonin.

Methods:- Mention method of randomization.

Results:- Mention p values for baseline data.

Discussion:- Emphasize the differences from other similar studies.

Conclusions:- Are justified on the basis of results of the study.

References:- Follow the format of the journal.

General:- Revise language, grammar and syntax.

6. PLOS authors have the option to publish the peer review history of their article (what does this mean?). If published, this will include your full peer review and any attached files.

Reviewer #1: No

Reviewer #2: **Yes: **Thomas Falconer

Reviewer #3: No

---

## [Author Response · Author response to Decision Letter 0]

29 Mar 2023

Journal Requirements:

- Done.

- 10.1007/s00228-021-03234-6. Epub 2021 Oct 20. 

In your revision ensure you cite all your sources (including your own works), and quote or rephrase any duplicated text outside the methods section. Further consideration is dependent on these concerns being addressed.

- I have addressed and rephrased the overlappning text.

-So sorry about that! It is corrected.

"LS, LM, YB, HJP have no conflict of interest. ME reports employment and stock ownership with SOBI AB, and previously with Vifor Pharma AB."

- The statement is included in the cover letter

- I have reviewed the reference list.

Additional Editor Comments:

Please respond to all reviewers comments

Reviewers' comments:

Reviewer's Responses to Questions

Comments to the Author

1. Is the manuscript technically sound, and do the data support the conclusions?

Reviewer #1: Yes

Reviewer #2: Yes

Reviewer #3: Yes

2. Has the statistical analysis been performed appropriately and rigorously? 

Reviewer #1: Yes

Reviewer #2: Yes

Reviewer #3: Yes

3. Have the authors made all data underlying the findings in their manuscript fully available?

Reviewer #1: No – DOI is now included

Reviewer #2: No - DOI is now included

Reviewer #3: Yes

4. Is the manuscript presented in an intelligible fashion and written in standard English?

Reviewer #1: Yes

Reviewer #2: Yes

Reviewer #3: Yes

5. Review Comments to the Author

Reviewer #1: A randomized-controlled clinical trial compared the level of pain between melatonin treatment and placebo in women with endometriosis-associated pain. No statistical differences in pain levels between the arms were observed.

Minor revisions:

1- Line 113: State the statistical testing method that achieves 80% power. If an estimate of the standard deviation was used in the power calculation, state it.

-We have clarified the power calculation and specified the assumed standard deviation that we used. The power calculation was based on independent t-test and we have clarified this by adding a reference to the program that we used for the calculation. 

2- Line 127: The standard statistical term for average is mean. – Changed

3- Table 1: Indicate if the summarized values in the table are means and standard deviations. If data is normally distributed, means and standard deviations are appropriate. However, if the data is non-normally distributed median, first and third quartiles are typically provided. – We specified that the numbers are mean values.

4- Line 148: Indicate the statistical testing method(s) used to conclude that the variables in table 1 were similar/different when comparing the groups. – It is now specified in the statistics section, we used independent t -test.

5- Line 149-156: In addition to the frequencies, provide the corresponding percentages. – Added

6- Line 161: Consider rephrasing the following sentence. Both the terms differences and similar were used which is confusing to the reader. “No statistically significant interaction between time and treatment group was seen, which imply that differences between the groups were similar for all time points.” – Rephrased to a, hopefully, more comprehensive phrase.

7- Line 163: State the percent with insomnia in each arm. – Added.

8- Define the abbreviation SD at its first occurrence. - Added.

Reviewer #2: This is a well thought out experiment. I agree with the authors that the context of the experiment allows for high internal validity. It is clear that the authors are aware of both the strengths and the weaknesses of the experiment.

• I think implementing a stratified random sampling to account for previous hormone therapy users and any other important confounders, ie, the number of amenorrhea patients recruited after the exclusion criteria was relaxed, would benefit the analyses and interpretation of the data.

-We agree, a sensitivity analysis stratified by hormonal treatment is added as a supplementary table. However due to the small number of observations it is hard to interpret.

• It also would be helpful to know when the patients in both arms conducted their self-assessments, since it is stated that the half life of melatonin is somewhere between 20 and 60 minutes. 

– We clarified that the self-assessment was also made at bedtime.

• For that matter, knowing the within-patient variance of documentation of self-assessments would be beneficial to the strength of the study. 

- We agree but unfortunately the data does not provide that information.

• The authors need to better define and describe the interaction terms used in the regression models. And disclosure of the model equation, including estimates and SEs, should be done before final acceptance. 

-We agree. In the statistical method section, we have now clarified which model we used in order to make inferens about the treatment difference between the groups. And which variables that were used in the models, when the interactions were added. 

It is my judgment that this experiment adds value to the scientific conversation regarding the association between melatonin and EAPP.

Reviewer #3: Thank you for giving me the opportunity to review this interesting RCT "Adjuvant use of melatonin for pain management in endometriosis-associated pelvic pain".

However, I have some comments and recommendations:

Title:- Is appropriate for the content of the article. – Thank you

Abstract:- Represents a suitable summary of the study. -Thank you

Introduction:- Emphasize the mechanism of action of melatonin. – We elaborated on the role of melatonin

Methods:- Mention method of randomization. – the very last section of method explains the randomization

Results:- Mention p values for baseline data. - Added

Discussion:- Emphasize the differences from other similar studies. - We clarified some differences

Conclusions:- Are justified on the basis of results of the study. ¬– Thank you

References:- Follow the format of the journal. – We changed the format accordingly

General:- Revise language, grammar and syntax. - done

6. PLOS authors have the option to publish the peer review history of their article (what does this mean?). If published, this will include your full peer review and any attached files.

Do you want your identity to be public for this peer review? For information about this choice, including consent withdrawal, please see our Privacy Policy.

Reviewer #1: No

Reviewer #2: Yes: Thomas Falconer

Reviewer #3: No

---

## [Decision Letter · Decision Letter 1]

10 Apr 2023

PONE-D-22-34783R1Adjuvant use of melatonin for pain management in endometriosis-associated pelvic pain - a randomized double-blinded, placebo-controlled trialPLOS ONE

Dear Dr. Söderman,

Thank you for submitting your manuscript to PLOS ONE. After careful consideration, we feel that it has merit but does not fully meet PLOS ONE’s publication criteria as it currently stands. Therefore, we invite you to submit a revised version of the manuscript that addresses the points raised during the review process.

Please respond to all reviewers comments

We look forward to receiving your revised manuscript.

Kind regards,

Ahmed Mohamed Maged, MD

Academic Editor

PLOS ONE

Journal Requirements:

Reviewers' comments:

Reviewer's Responses to Questions

**Comments to the Author**

1. If the authors have adequately addressed your comments raised in a previous round of review and you feel that this manuscript is now acceptable for publication, you may indicate that here to bypass the “Comments to the Author” section, enter your conflict of interest statement in the “Confidential to Editor” section, and submit your "Accept" recommendation.

Reviewer #1: (No Response)

Reviewer #3: All comments have been addressed

2. Is the manuscript technically sound, and do the data support the conclusions?

Reviewer #1: Yes

Reviewer #3: Yes

3. Has the statistical analysis been performed appropriately and rigorously? 

Reviewer #1: Yes

Reviewer #3: Yes

4. Have the authors made all data underlying the findings in their manuscript fully available?

Reviewer #1: Yes

Reviewer #3: Yes

5. Is the manuscript presented in an intelligible fashion and written in standard English?

Reviewer #1: Yes

Reviewer #3: Yes

6. Review Comments to the Author

Reviewer #1: Minor revision:

1- Abstract: Include the corresponding standard deviations for 2.9 and 3.3.

2- Line 118: State that the independent t-test was the statistical method used for estimating the power of the study.

3- Line 155: For improved clarity consider replacing the sentence with the following. "The outcomes are presented as means and SD of the screen cycle."

Reviewer #3: Thank you for giving me the opportunity to review this interesting RCT "Adjuvant use of melatonin for pain management in endometriosis-associated pelvic pain".

Almost all reviewer’s comments and recommendations were considered by the authors.

7. PLOS authors have the option to publish the peer review history of their article (what does this mean?). If published, this will include your full peer review and any attached files.

Reviewer #1: No

Reviewer #3: **Yes: **Akmal El-Mazny

---

## [Author Response · Author response to Decision Letter 1]

15 Apr 2023

Journal Requirements:

The reference list is revised and is accurate. No changes have been made.

Reviewer #1: Minor revision:

1- Abstract: Include the corresponding standard deviations for 2.9 and 3.3. - Included

2- Line 118: State that the independent t-test was the statistical method used for estimating the power of the study. – the addition is made.

3- Line 155: For improved clarity consider replacing the sentence with the following. "The outcomes are presented as means and SD of the screen cycle." – I added SD.

---

## [Decision Letter · Decision Letter 2]

11 May 2023

Adjuvant use of melatonin for pain management in endometriosis-associated pelvic pain - a randomized double-blinded, placebo-controlled trial

PONE-D-22-34783R2

Dear Dr. Söderman,

We’re pleased to inform you that your manuscript has been judged scientifically suitable for publication and will be formally accepted for publication once it meets all outstanding technical requirements.

Kind regards,

Ahmed Mohamed Maged, MD

Academic Editor

PLOS ONE

Additional Editor Comments (optional):

Reviewers' comments:

Reviewer's Responses to Questions

**Comments to the Author**

1. If the authors have adequately addressed your comments raised in a previous round of review and you feel that this manuscript is now acceptable for publication, you may indicate that here to bypass the “Comments to the Author” section, enter your conflict of interest statement in the “Confidential to Editor” section, and submit your "Accept" recommendation.

Reviewer #1: All comments have been addressed

2. Is the manuscript technically sound, and do the data support the conclusions?

Reviewer #1: (No Response)

3. Has the statistical analysis been performed appropriately and rigorously? 

Reviewer #1: (No Response)

4. Have the authors made all data underlying the findings in their manuscript fully available?

Reviewer #1: (No Response)

5. Is the manuscript presented in an intelligible fashion and written in standard English?

Reviewer #1: (No Response)

6. Review Comments to the Author

Reviewer #1: (No Response)

7. PLOS authors have the option to publish the peer review history of their article (what does this mean?). If published, this will include your full peer review and any attached files.

Reviewer #1: No

---

## [Editor Report · Acceptance letter]

26 May 2023

PONE-D-22-34783R2 

Adjuvant use of melatonin for pain management in endometriosis-associated pelvic pain - a randomized double-blinded, placebo-controlled trial 

Dear Dr. Söderman:

I'm pleased to inform you that your manuscript has been deemed suitable for publication in PLOS ONE. Congratulations! Your manuscript is now with our production department. 

Kind regards, 

on behalf of

Professor Ahmed Mohamed Maged 

Academic Editor

PLOS ONE